# Unprecedented decline of Arctic sea ice outflow in 2018

Hiroshi Sumata 1✉, Laura de Steur 1, Sebastian Gerland1, Dmitry V. Divine1 & Olga Pavlova1

Fram Strait is the major gateway connecting the Arctic Ocean and North Atlantic Ocean, where nearly 90% of the sea ice export from the Arctic Ocean takes place. The exported sea ice is a large source of freshwater to the Nordic Seas and Subpolar North Atlantic, thereby preconditioning European climate and deep water formation in the North Atlantic Ocean. Here we show that in 2018, the ice export through Fram Strait showed an unprecedented decline since the early 1990s. The 2018 ice export was reduced to less than 40% relative to that between 2000 and 2017. The minimum export is attributed to regional sea ice-ocean processes driven by an anomalous atmospheric circulation over the Atlantic sector of the Arctic. The result indicates that a drastic change of the Arctic sea ice outflow and its environmental consequences happen not only through Arctic-wide ice thinning, but also by regional scale atmospheric anomalies.

---

1 Norwegian Polar Institute, Fram Centre, Tromsø, Norway. ✉email: hiroshi.sumata@npolar.no

Sea ice export through the Fram Strait plays a central role in the freshwater cycle connecting the Arctic and subarctic regions in the Northern Hemisphere[1,2]. Since the Arctic Ocean gathers 11% of the global river discharge into 3% area of the global ocean[3], sea ice export, together with low salinity sea-water carried by ocean currents, returns excess of the freshwater inputs toward lower latitudes[4]. The freshwater export mainly occurs in the Atlantic sector of the Arctic, thereby setting up surface stratification of the neighboring Nordic Seas and further downstream. The freshwater delivered to the Nordic Seas, either in liquid form or as sea ice, preconditions the physical environment for marine ecosystems[5,6], European weather, and climate[7,8], and modulates the rate of deep water formation in the downstream North Atlantic Ocean with implications for climate on a global scale[9–11].

Sea ice export through Fram Strait accounts for ~90% of the total sea ice outflow from the Arctic[12,13]. In the 20th century, the amount of ice export was equivalent to the liquid freshwater transport in the East Greenland Current (EGC)[12,13]. The sea ice export, however, has shown a reduction in the last decades, mainly attributed to a reduction of mean ice thickness in Fram Strait over the last two decades at a rate of ~15% per decade[14]. Climate simulations point to a continued decline in the coming decades with direct consequences for the hydrological cycle in the Arctic and Subarctic regions[15,16].

Sea ice export and liquid freshwater export with the EGC in the western half of Fram Strait extends approximately from the Prime Meridian to the continental shelf of East Greenland with strong seasonal fluctuations[17,18] (Fig. 1). Within and just north of the strait, the EGC meets returning warm and saline Atlantic Water (AW) that circulates westward following several pathways between that are typically associated with large eddy variability[19,20]. The recirculating AW meets and may subduct under the fresher Polar Water, and flow southward again along the Polar front with the EGC. In 2018, AW was found as far north as 82.8°N west of the Yermak Plateau[21].

Sea ice in Fram Strait is originally formed upstream in the Arctic Ocean. The major part of this ice originates from the Siberian shelves (i.e., the Laptev Sea, the East Siberian Sea and to a lesser extent, the Chukchi Sea) and the southern perimeter of the perennial sea ice cover in the central Arctic[22,23]. The ice floes experience seasonal thickening and thinning during their advective pathway toward the Fram Strait. The Transpolar Drift stream, driven by an anticyclonic wind field centered in the Canada Basin, is the main mechanism transporting ice floes from the Siberian shelves to Fram Strait. Ice floes in Fram Strait continue their southward motion along with the EGC, partially driven by northerly wind associated with a sea-level pressure (SLP) gradient between Greenland and the Barents Sea[24,25]. A perennial difference of surface temperature between northern Greenland and the Barents Sea maintains this pressure gradient, setting up northerly winds and resulting in southward sea ice motion through Fram Strait year-round, though most prominent in winter[26].

Here we report a strong decline in sea ice export through Fram Strait in 2018, unprecedented since observations started in the 1990s, and evidenced by ice thickness measurements from Upward Looking Sonar (ULS) combined with satellite remote sensing observations. The cause of this decline is found through an analysis combining backward trajectories of ice floes, remote sensing products, and atmospheric reanalysis data.

## Results and discussion

**Sea ice volume transport through Fram Strait.** Sea ice volume export is calculated from ice thickness, an areal fraction of sea ice (sea ice concentration), and ice drift speed. Passive microwave satellite observations have provided the fraction of sea ice and ice drift speed covering the entire Arctic and Subarctic for all seasons since 1979[27,28], whereas ice thickness estimates from satellites are still limited to the freezing season due to properties of the ice surface in the melting season[29]. Upward Looking Sonar (ULS)

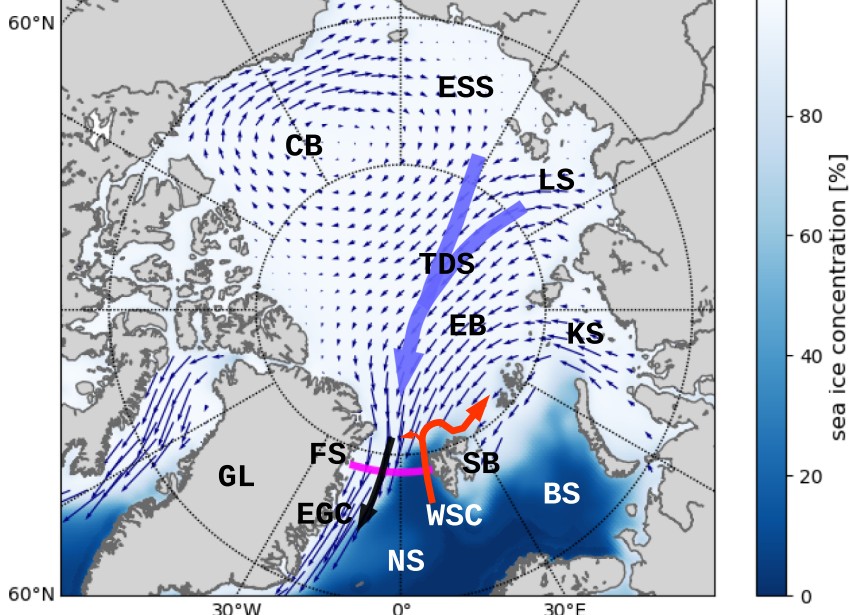

**Fig. 1 Sketch of the Arctic Ocean and subarctic seas with sea ice drift pattern and ocean currents through Fram Strait.** BS Barents Sea, CB Canada Basin, EB Eurasian Basin, EGC East Greenland Current (thick black arrow), ESS East Siberian Shelf, FS Fram Strait (thick magenta line), GL Greenland, KS Kara Sea, LS Laptev Sea, NS Nordic Seas, SB Svalbard, TDS transpolar drift stream (thick sky blue arrow), WSC West Spitsbergen Current (thick orange arrow). The background color shows mean sea ice concentration in winter (January–March, 1980–2018 mean), and small blue arrows show annual mean sea ice drift field in 1980–2018.

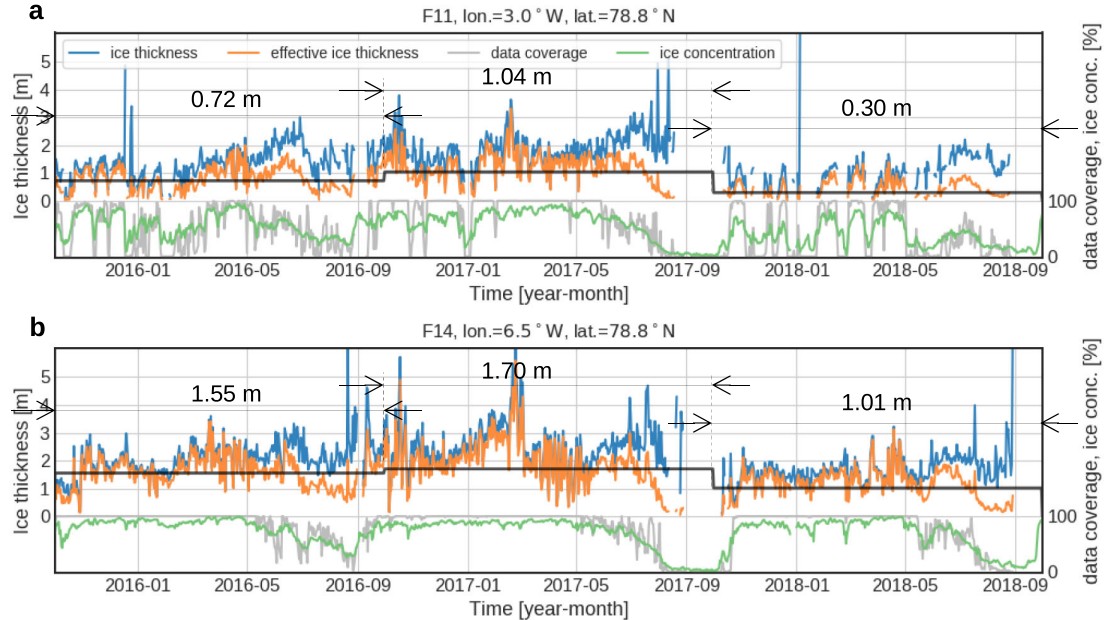

**Fig. 2 Daily time series of sea ice properties in Fram Strait for 2016–2018 period. a** Easternmost site F11 and **b** westernmost site F14. The black lines depict annual mean effective ice thickness in each year, the period and value of which are shown by arrows and a number. The annual mean values are defined by an average from October of preceding year to September of the year. A longer time series (2003–2018) of daily sea ice properties of all moorings (F11–F14) can be found in Fig. S14 in the supplementary material.

installed underneath sea ice, however, can provide accurate, near-continuous measurements of ice thickness covering all seasons[22,26]. The Fram Strait Arctic Outflow Observatory (AOO) maintained by the Norwegian Polar Institute has been monitoring ice thickness and ocean properties in the EGC at about 79°N in the western Fram Strait since the early 1990s. An array of four ocean moorings at ~3.25°W, 4°W, 5°W, and 6.5°W has been equipped with ULSs covering the main tongue of sea ice in the EGC. The observatory has documented long-term estimates, variability, and changes of the freshwater, and sea ice export from the Arctic Ocean to the Nordic Seas[14,30].

In 2018, the ULS array recorded an unprecedented decline of ice draft (from which ice thickness is calculated) since the comprehensive measurements started in 1990. A sudden drop of ice thickness was recorded by all four ULSs starting in August 2017 and lasting until the end of the record in August 2018 (Fig. 2). Here we refer to 'annual mean' as the average from September to August in order to cover the full seasonal cycle of sea ice. The decline in 2018 was accompanied by a reduction of sea ice concentration particularly evident in the eastern part of the EGC (Fig. 2, green lines). The measured drop in the ice thickness, together with the reduction of ice concentration, resulted in a large reduction of sea ice volume in Fram Strait. The reduction of annual mean effective ice thickness (this is the thickness weighted for ice concentration across the array) was 0.74 m relative to the preceding year (average of the four mooring sites, see also numbers embedded in Fig. 2), corresponding to a reduction of ice volume by 55% in this area (3°–6.5°W at 78.8°N).

As a direct consequence of the large reduction in ice volume, annual sea ice volume transport reached a record minimum in 2018 since the observations started (Fig. 3c). The annual volume transport in 2018 was just 590 km$^3$ yr$^{-1}$, implying a reduction of 60% relative to the 2000–2017 period and 75% relative to the 1990s. (Table 1). The minimum fell below the prior minimum recorded in 2013 (following the record minimum of the Arctic summer sea ice extent in 2012) by 36%, indicating the anomalous

situation in 2018 is exceptional. The decline was compounded by a concurrent slowdown of ice drift speed (Fig. 3b). Southward sea ice motion was particularly low from January to July in 2018 (66% slower than 1990–2018 average), resulting in the weakest annual mean drift speed since 1992. The weak drift was found to be co-responsible for the observed thickness decline through atmospheric conditions, as we will show in the following sections.

**Origin and modification of sea ice north of Fram Strait.** To investigate the cause of the ice thickness decline in Fram Strait, we constructed backward trajectories of sea ice floes in Fram Strait. The temporal evolution of ice thickness is examined along with ice drift trajectory back in time, together with concurrent thermo-dynamic forcing on the ice floes (see method section). Figure 4a shows backward trajectories of ice floes that arrived in Fram Strait in May each year from 2011 to 2017 and 2018. The trajectories describe the origins and pathways of the ice floes (lines in Fig. 4a) along with their locations three months before arrival to the strait (diamonds in Fig. 4a). The ice floes in 2018 did not show very different pathways compared to those of the preceding years 2011–2017, indicating that neither the origins nor pathways were the cause of the anomalous thickness in 2018. Figure 4b shows the temporal evolution of ice thickness along the trajectories shown in Fig. 4a. When looking at the thickness of sea ice upstream more than 1 year prior to arriving at Fram Strait in 2018, the sea ice was notably thicker than the ice in 2011–2017 (Fig. 4b preceding months from −30 to −13). Thickness of the sea ice arriving in Fram Strait in 2018, however, started to decline rapidly three months before arriving at the strait (Fig. 4b preceding months −3 to −1). This was unusual compared with the thickness evolutions of the preceding years that exhibited further thickening during this period in mid-winter (February–March) (orange vs. blue lines in Fig. 4b at preceding months from −3 to −2).

Associated with the 2018 trajectories, we find anomalously warm atmospheric conditions during the preceding winter

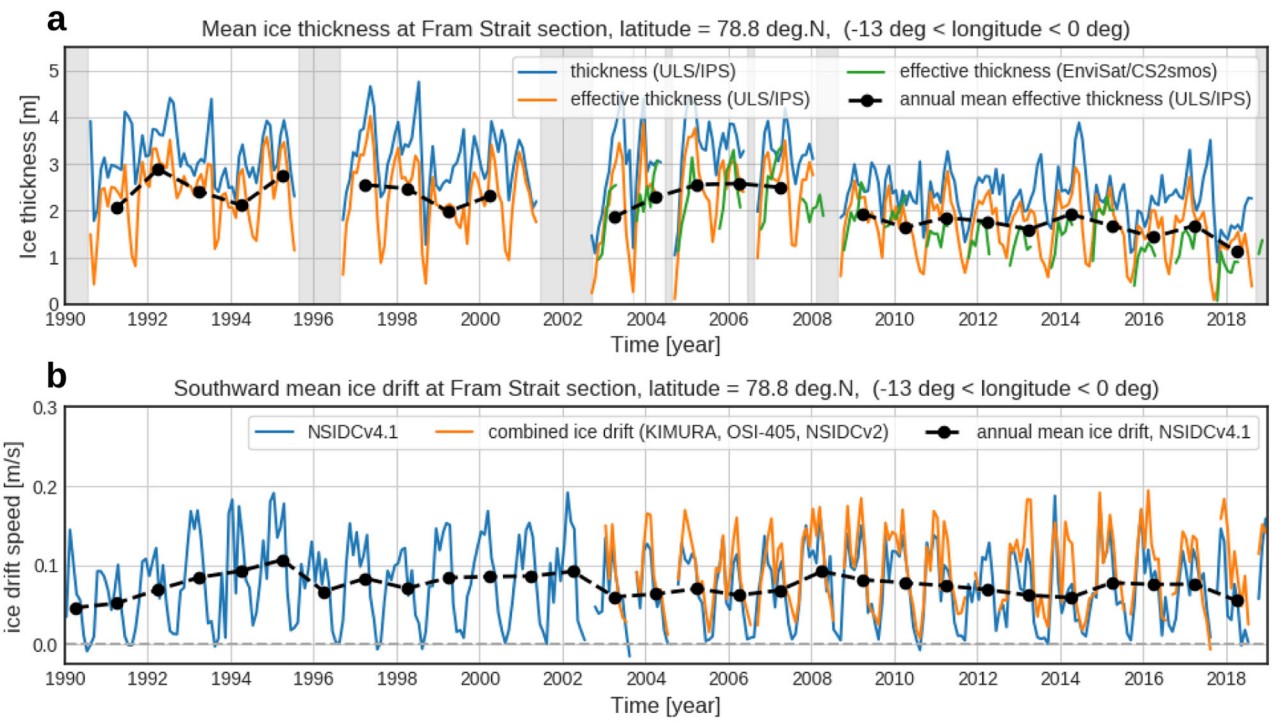

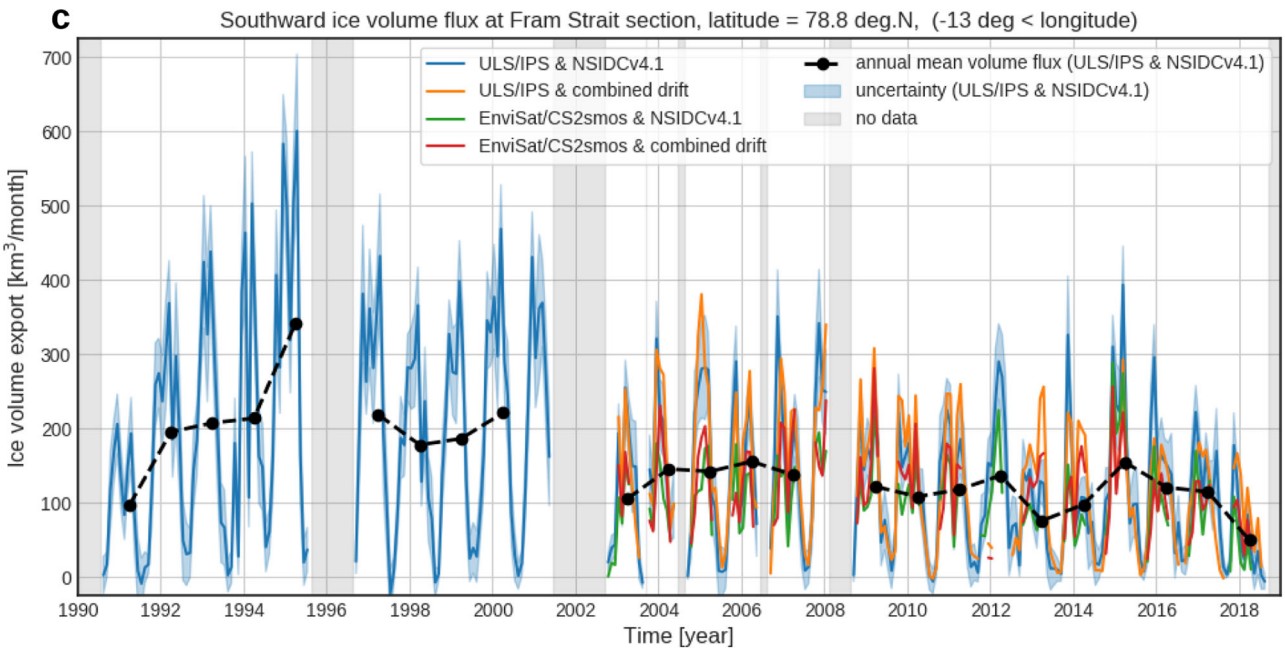

**Fig. 3 Monthly mean time series of sea ice properties at the Fram Strait section for 1990–2018. a** Ice thickness, **b** southward ice drift speed, and **c** southward ice volume transport. Estimates from different data sources are shown in different color. Annual mean values are calculated from averages from October of preceding year to September of the year to represent the seasonal cycles of sea ice formation and melt. See method section for detail. AWI CS2smos product is used in the thickness time series in a and southward volume transport in **c**.

months (Oct. 2017–Feb. 2018 or preceding months −8 to −3) at the locations where the ice pack were then, before arriving to Fram Strait (Fig. 4c). The warm conditions showed a relative maximum in February 2018, coinciding with the start of the anomalous sea ice thinning in 2018 (Fig. 4b). In February 2018, monthly mean 2 m air temperatures along the trajectories were nearly 10 degrees higher than the average of the preceding years 2011–2017. While these warm conditions likely have decelerated further thickening of the ice floes in comparison with the

preceding years, it is unlikely that the atmospheric heat supply caused the drastic thinning of the ice floes during February–April 2018. The highest monthly mean 2 m air temperature in this period was still below −10 °C, i.e., well below the freezing point of seawater.

The other prominent difference associated with the trajectories in 2018 relative to 2011–2017, is the location of the ice floes 3 months before arriving at the strait (diamonds in Fig. 4a and Fig. 4d). In 2018, the ice floes stayed in an area north of Fram

**Table 1 Estimates of southward annual sea ice volume flux at the Fram Strait section.**

| Period [year] | Volume flux [km³ yr⁻¹] | |
|---|---|---|
| | Mean ± uncertainty[a] | ±std. dev. |
| 1990–1999 | 2450 ± 650 | ±760 |
| 2000–2009 | 1760 ± 470 | ±410 |
| 2010–2017 | 1390 ± 430 | ±270 |
| 2018 | 590 ± 260 | — |

[a]The uncertainties refer to those of annual mean volume flux in each period and in 2018. They are calculated from uncertainties of monthly mean volume flux shown in Fig. 3c by taking propagation of error into account. The standard deviations also refer to those of the annual mean volume flux. See description in method section.

Strait about 200–300 km closer to that compared to the ice floes in 2011–2017. This area north of Fram Strait is characterized by a strong heat supply from the underlying AW to sea ice[31–33]. The trajectory analysis shows that the ice floes were thinning rapidly between 82°N (~300 km north of Fram Strait) and the mooring array at ~79°N (Figs. S1–S6, summarized in Table S1). In 2011–2017, ice floes reached this area approximately 1 month before the arrival at the strait and experienced a substantial melting (Table S1). According to the ice thickness changes along the 2011–2017 trajectories, the average melt rate during the last month before arriving at the strait is 0.8 cm day⁻¹ in winter (December–February) and 1.7 cm day⁻¹ in early spring (March–April) (Table S2), which is consistent with the melt rates of 0.3–9 cm day⁻¹ in spring obtained from in situ measurements north of Fram Strait[34]. In 2018, the sea ice reached the area south of 82°N already three months before arriving at Fram Strait, allowing for the ice floes to be exposed to strong heat supply from the ocean two to three times longer than in the preceding years. The additional reduction in ice thickness associated with this longer residence time is estimated to be 20–50 cm in winter and 50–100 cm in spring. These numbers quantitatively explain the acceleration in thickness decline observed in winter 2018 (Fig. 4b and Figs. S1–S6).

**Anomalous sea-level pressure pattern in 2018**. The longer residence time of the ice floes in the northern vicinity of Fram Strait was a consequence of slow and stagnant ice motion from autumn 2017 to summer 2018. The stagnation was caused by a repeated reversal of the SLP gradient across the strait and associated changes in prevailing wind direction during this period (Fig. 5c, d). An east–west dipole SLP anomaly repeatedly emerged in the Atlantic sector of the Arctic Ocean, most prominent in February 2018 (Fig. 5a, b). This pattern has also been recognized for the unusual emergence of the polynya north of Greenland in 2018[35,36]. The spatial pattern of these dipole anomalies is characterized by a high-pressure center in the Barents Sea and a low-pressure center over the Canadian Archipelago and Greenland.

These anomaly patterns are different from the 'Arctic dipole anomaly' with respect to the anomaly centers. The Arctic dipole anomaly is a characteristic pressure anomaly pattern over the Arctic region, having its one of the centers of anomaly in Kara Sea and Laptev Sea[37]. The Arctic dipole anomaly was reported to influence the sea ice export through Fram Strait in winter[38]. The SLP anomalies that occurred in the 2017–2018 period had a more direct impact on the ice export compared to the Arctic dipole anomaly pattern, since they were situated such that Fram Strait was sandwiched between the anomaly centers. The reversals of zonal SLP gradient across Fram Strait led to concurrent southerly winds that reduced, and even halted, southward sea ice motion in Fram Strait (Fig. 5d, Figs. S3–S6 bottom panels). The spatial

pattern of the anomalies is very similar to the earlier reported SLP anomaly pattern associated with variations in sea ice motion in Fram Strait[25] (i.e., the second leading empirical orthogonal function (EOF) mode of daily SLP in the Atlantic sector (45–90°N, 90°W–90°E)). Though the EOF mode was derived from daily SLP analysis, the 2018 dipolar anomalies emerged repeatedly on a monthly time scale with a slightly different spatial pattern (Fig. S7).

The southerly wind events reduced the annual mean southward motion of sea ice to Fram Strait significantly and resulted in enhanced ice melt due to the longer residence time north of the strait where AW meets the transpolar drift stream. At the same time, the southerly winds carried warm air masses to the area north of Fram Strait, leading to anomalous warm air temperatures in this area, which also had a negative impact on ice growth in winter (Fig. 5e).

**Discussions and wider implications**. This unprecedented reduction of sea ice volume export by nearly 60% relative to 2000–2017 and 75% relative to the 1990s average, can have a large impact on atmosphere–sea ice–ocean processes, ocean density contrasts, and consequently climate both in the Arctic Ocean and downstream North Atlantic Ocean. A salinity section in the Fram Strait in May 2018 shows fresher conditions in the surface layer relative to observations from spring 2002, 2005, 2007, and 2008, concurrent with the sea ice melt in the upstream area (Fig. S13a, b). However, the 2018 salinity anomaly in the upper ocean of −0.35 psu (0–30 m average, Fig. S13c and Table S4), accounts for less than one-third of a potential salinity change due to meltwater input associated with the ice thickness anomaly in this period, i.e., 1.22 m ice thickness reduction can cause salinity anomaly of −1.21 psu (0–30 m average, Table S4). This suggests that a large part of the meltwater remains in the upstream area of Fram Strait, is mixed down, or is exported at least not at the same time. Sea ice melt in the northern Fram Strait increases surface buoyancy in this part of the Arctic, while further south along the EGC front, buoyancy is reduced due to less southward ice transport[39], allowing for more direct cooling of Atlantic-origin water in winter[40].

The difference of the fate of sea ice export versus oceanic transport results in different consequences depending on the phase of the freshwater passing through the Fram Strait. A significant portion of sea ice passing through the Fram Strait can drift into the Nordic Seas[41,42]. The liquid freshwater transport on the shelf and in the EGC, however, will follow mostly the continental shelf break of Greenland with some loss through the Jan Mayen Current and East Icelandic Current[43,44]. Therefore, a liquid freshwater anomaly due to enhanced ice melting upstream of the Fram Strait may propagate further south with the EGC, with a potential consequence for the dense water formation in the North Atlantic Ocean[45]. In addition, when sea ice volume carried with the EGC is reduced, a further retreat of the sea ice edge in the Greenland and Iceland Seas occurs[39]. This allows for stronger air–sea interactions and water mass modification along the ice edge of the EGC in winter[40] and modulates primary production levels in spring[39]. The aforementioned impacts on downstream stresses the need for comprehensive analysis of the data corrected by the ocean observatories like the Fram Strait AOO, and major observational arrays further south in the Atlantic Ocean (e.g., OSNAP[46], RAPID-MOCHA[47]) to quantify the potential effects of significantly reduced sea ice export.

It is also important to note the consequences of the anomalous thickness reduction and the southerly wind events on the local hydrography, water mass modification, and biological processes north of the Fram Strait. A recent study reported that anomalous reduction of sea ice concentration and extreme deep mixed layers

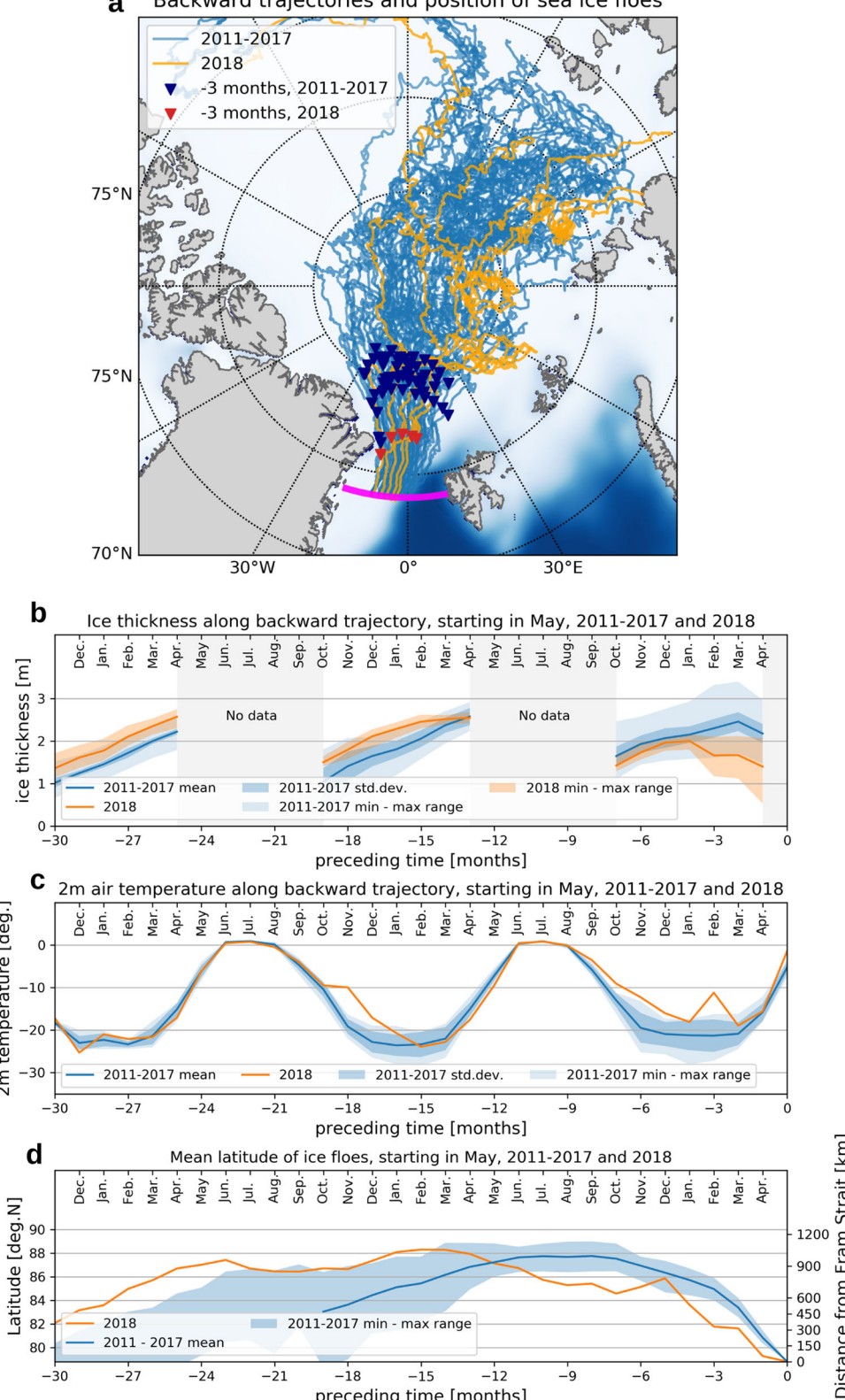

**Fig. 4 Backward trajectory map and reconstructions of associated physical properties. a** trajectories of sea ice floes which arrived at Fram Strait in May, 2011–2017 and 2018, and reconstructions of **b** ice thickness, **c** 2 m air temperature, and **d** latitude along the trajectories back in time. The magenta line in panel **a** shows Fram Strait. The right edge of the panels **b**–**d** denotes arrivals of the ice floes at Fram Strait, while the left edge denotes 2.5 year (30 months) before the arrival. Corresponding months are shown at the top of the panels **b**–**d**. The ice thickness in panel **b** shows an average of independent reconstructions by three different ice thickness products (AWI CS2smos, Univ. Bristol, and NERC-CPOM) derived from CS2smos/CS2. The 2 m air temperature in panel **c** is obtained from ERA5. See method section for a description.

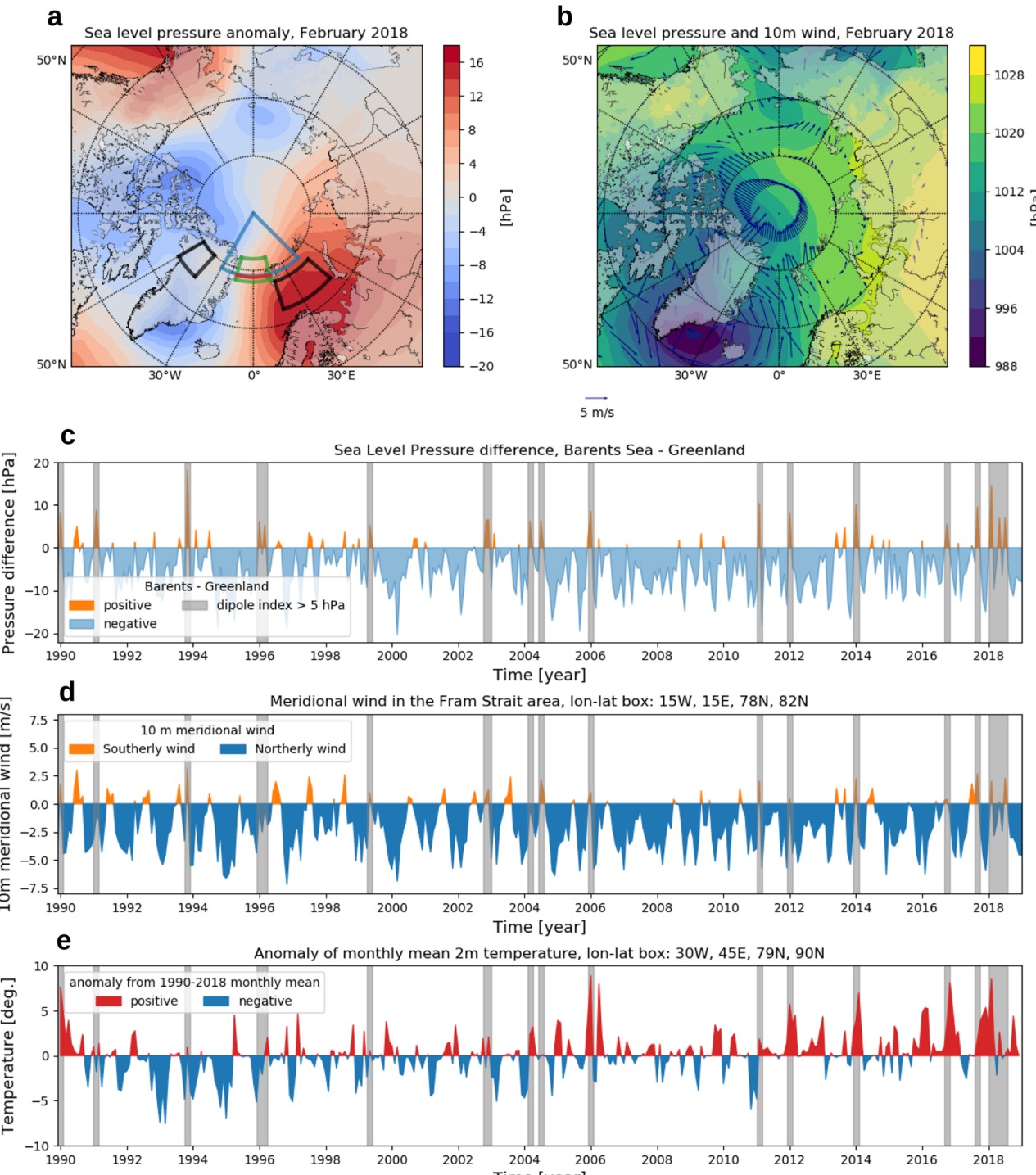

**Fig. 5 Spatial pattern and time series of sea-level pressure anomaly, wind field, and 2 m air temperature over the Fram Strait area from European Center for Medium-Range Weather Forecasts Reanalysis v5 (ERA5). a** Sea-level pressure anomaly in February, 2018 relative to 1990–2018 mean, **b** wind and pressure field in February, 2018, time series of **c** sea-level pressure difference between the Barents Sea and Greenland, **d** 10 m meridional wind over the Fram Strait area, and **e** 2 m air temperature anomaly in the north of Fram Strait. The pressure difference is calculated from the averages in the two black polygons shown in panel **a**, used in Tsukernik et al. (2010)[24]. The 10 m meridional wind and 2 m air temperature anomaly are calculated by an average in the green polygon and the blue sector in panel **a**. The gray shades shown in panels **c**, **d**, and **e** depict positive pressure difference (Barents Sea—Greenland) larger than 5 hPa on monthly basis.

occurred northeast of Fram Strait in the 2017/2018 winter[48]. Another study pointed out that the anomalous ocean stratification in 2018 affected local biogeochemical cycles north of the Fram Strait[49]. The anomalous ice melt reported here can have contributed to the reduction of sea ice cover in this area, while the

southerly winds can have advected the meltwater eastward by Ekman transport and contributed to the observed fresher surface layer and shallower warm AW layer observed in the western Eurasian basin in winter 2017[21]. The anomalous thickness reduction, together with the observed hydrographic anomalies

in the northeast of Fram Strait, might be a leading signal of further retreat of marginal ice zones and concurrent northward shift of water mass modification area in the Atlantic sector of the Arctic in the coming years to decades. Sustained monitoring efforts of both sea ice and upper ocean properties in the Atlantic sector of the Arctic Ocean are necessary to clarify both short-term and long-term impacts of the 2018 event for the Arctic Ocean itself.

The effect of a preconditioned environment and potential recurrences of such southerly wind events in a changing climate also need to be considered. More heat transport into the Arctic Ocean[50,51] might result in an increase of ocean heat content available for melting sea ice, a decline of basin-wide ice thickness[52] making ice floes more vulnerable to environmental changes, an increase of mobility of sea ice in a number of regions[53] making sea ice more sensitive to local wind forcing; all these might have preconditioned the 2018 ice export decline. Climate model simulations suggest that the freshwater cycle of the Arctic may shift into a completely different regime in the next decades, with a consequence of reduced ice export and increased liquid fraction of freshwater transport[16]. Hence, a repeated appearance and the cause of the dipole SLP anomaly like in 2018, and the likelihood of recurrences in the future also need further investigation in this context. A relation between the most prominent anomaly in February 2018 and sudden stratospheric warming has been pointed out[35], whereas the cause of the repeated occurrence of similar patterns in 2017–2018, its relation to dominant atmospheric patterns such as North Atlantic Oscillation or North Atlantic blocking activity, and potential occurrence in a changing climate is still unclear. Interactions and feedbacks between these components in the freshwater cycle and resulting impacts on the ocean ecosystem and European climate are ongoing challenges.

## Methods

**Data**. Ice thickness data were obtained from Upward Looking Sonars (ULSs) and Ice Profiling Sonars (IPSs) moored in the East Greenland Current in Fram Strait. Four ULS/IPSs were zonally aligned at 79°N from 1990 to 2001 and at 78.8°N from 2002 to 2018 with two temporal gaps in 1996 and 2008 and with several spatial gaps in between. Zonal positions (names) of the moorings are 3°W (F11), 4°W (F12), 5°W (F13), and 6.5–7.0°W (F14), respectively (Fig. S8). The moorings are serviced annually. Locations and temporal coverages of the ULS/IPSs array until 2014 were described in past literature[14]. After 2014, four IPSs were continuously operated except for two data gaps (missing F13 in 2014 and F12 in 2015). Although the zonal locations of the moorings have slightly changed from year to year, the changes are taken into account in the zonal interpolation procedure of the ice volume transport calculation.

The ULS and IPS measure the underwater fraction of sea ice (sea ice draft) by recording travel times of sound reflected at the bottom of floating sea ice[54]. The raw data were processed to ice draft data by ASL Environmental Science Inc., following standard procedures[54,55] including screening of erroneous records (e.g., despiking) and sound speed corrections[56]. The ice draft is converted to ice thickness by an average ratio of thickness to draft in Fram Strait, 1.136, derived from on-site drilling measurements[57]. Ice draft measurement interval varies with the instruments used and their respective settings. From 1990 to 2005, ULSs (ES300) were used to obtain the draft data, measurement intervals of which vary from 60 to 600 seconds, though in most cases an interval of 240 seconds was employed. After 2006, IPSs (IPS4 and IPS5) were used to measure the ice draft. The measurement interval of IPS4 and IPS5 was set to two seconds, though an interval of one second was applied in some IPSs from 2014 to 2016.

Daily mean ice draft and thickness are calculated by averaging the original data except for the open water fraction. Hence the daily mean thickness exhibits a spiky time series when the area is largely occupied by open water (Fig. 2 blue lines). Effective ice thickness is then calculated by taking the fraction of open water (sea ice concentration) into account. Hence the effective thickness corresponds to ice volume averaged over the daily period. Monthly mean ice thickness and effective ice thickness are calculated by the daily thickness and effective thickness (Fig. 3a). Temporal coverage of each daily data gives weight in this mean calculation. The monthly mean effective ice thickness is used for the volume flux calculation.

We also apply altimetry-based sea ice thickness products from remote sensing for the backward trajectory analysis and validation of interannual variation of the ULS/IPS thickness time series in the freezing season (Fig. 3a, green line). The remote sensing products give thickness estimates with a basin-wide spatial coverage, although their temporal coverages are limited to the freezing season from October to April. It has been pointed out that the products differ in their absolute magnitude though they agree in the spatial distribution and basin-scale gradients[58,59]. To take into account the uncertainty of the thickness estimates from remote sensing, we apply three different thickness products for the trajectory analysis independently and use the mean and standard deviation of the results of the three applications as a measure of thickness estimates along the trajectories. The used products are, the weekly Arctic sea ice thickness derived from CryoSat and SMOS using an optimal interpolation scheme (hereafter AWI CS2smos product)[29], Arctic sea ice and physical oceanography derived from CryoSat-2 Baseline-C Level 1b waveform observations (hereafter Univ. Bristol thickness product)[60], and CryoSat Operational Polar Monitoring Product (hereafter NERC-CPOM thickness product)[61]. These three products provide ice thickness estimates based on CryoSat-2 radar altimeter measurements[62] combined with different thickness retrieval algorithms[59]. Two of the products (AWI CS2smos and Univ. Bristol) in addition merged ice thickness estimates from Soil Moisture and Ocean Salinity satellite (SMOS) to exploit its advantage in thin sea ice area[63]. The time period of the products used in this study is from 2010 to 2018. Ice thickness estimates by Envisat[64] are also used for validation of interannual variation of the ULS/IPS time series in the period 2002–2010 for winter months (Fig. 3a, green line).

Sea ice drift products derived from satellite remote sensing are used to calculate ice volume flux at the Fram Strait section and backward trajectories of ice floes. Polar Pathfinder Daily 25 km EASE-Grid Sea Ice Motion Vectors, version 4.1[28] (hereafter NSIDCv4 ice drift) is used for both the volume flux and the trajectory calculation. The product serves the longest temporal coverage (1979–2018) among available drift products by a combination of ice motion estimates derived from various sensors and tracks of ice tethered buoys, which suits both the volume flux and trajectory calculation for a long-range (Fig. 3b). Other drift products are also employed for validation purposes, although they have temporal gaps and temporal coverages are limited in recent years. To construct a seamless time series, we combine three different basin-wide products; OSI-405[65] and the sea ice motion estimates by Kimura[66] (hereafter Kimura ice drift), and the Polar Pathfinder Daily 25 km EASE-Grid Sea Ice Motion Vectors, version 2[67,68] (hereafter NSIDCv2 ice drift). The three products are combined to take their full advantage: OSI-405 is employed if data are available, KIMURA is used for deriving ice drift for summer months, and NSIDCv2 ice drift is used to fill temporal gaps of the two products. Due to the combination of the products in different quality, the merged ice drift has artificial temporal variations in basin-averaged drift speed[69]. Hence we carefully apply the merged ice drift for partial validation of NSIDCv4 ice drift time series after 2002 (Fig. 3b, orange line).

Sea ice concentration is taken from the Global Sea Ice Concentration Climate Data Records of the EUMETSAT OSI SAF[27] (OSI-409 version 1.2 for a period from January 1990 to April 2015, and OSI-430 version 1.2 for a period after April 2015). The products contain robust uncertainty estimates as well as sea ice concentration estimates. The sea ice concentration and its uncertainty are used to calculate sea ice volume flux at the Fram Strait section and to provide a termination condition of the backward trajectory calculation.

Atmospheric data are taken from European Center for Medium-Range Weather Forecasts Reanalysis v5[70] (ERA5). Daily and monthly mean 2 m air temperature, sea-level pressure, and 10 m wind are used in the analyses.

Salinity data in the western part of Fram Strait in April–May are obtained from shipboard CTD (conductivity temperature depth) measurements during the following cruises: Arctic Ocean 2002 (Oden), Fram2005 (RV Lance and KV Svalbard), iAOOS 2007 (KV Svalbard), iAOOS 2008 (KV Svalbard), and JR17005 (2018, RSS James Clark Ross).

**Sea ice volume transport in Fram Strait**. The Fram Strait section is discretized in 0.2 degrees (4.32 km at 78.8° N) zonal segments for the sea ice volume transport calculation. The volume transport at a segment is given by

$$f(T, A, V) = T(x) \cdot A(x) \cdot V(x), \tag{1}$$

where $T$, $A$, and $V$ are sea ice thickness, ice concentration, and ice drift velocity perpendicular to the section at location $x$, respectively. Sea ice concentration and ice drift speed at each segment are linearly interpolated from monthly mean values at the nearest data points. Monthly mean sea ice concentration is calculated from the daily sea ice concentration of OSI-409 and OSI-430, while monthly mean sea ice drift speed is calculated from daily values of NSIDCv4. Sea ice thickness at each segment is calculated from a linear interpolation of thickness between nearest mooring sites if the segment is located in the range of the mooring array (6.5° W < $x$ < 3.0° W). The validity of the interpolation is supported by high correlations of monthly mean values of ice thickness (0.64–0.94) and effective ice thickness (0.81–0.95) between the mooring sites (Fig. S9). If the segment is located outside the mooring array ($x$ < 6.5° W or $x$ > 3.0° W), an extrapolation is applied. The extrapolated thickness is calculated from ice thickness at the outermost mooring site and a mean zonal gradient of ice thickness in the Fram Strait section. Monthly mean values of the zonal (spatial) gradient, α are estimated from the ice thickness gradient between the four mooring sites for all observations between 2010 and 2018 (Fig. S10).

The total sea ice volume transport in Fram Strait, $F$, is given by a zonal integration of (1),

$$F = \int T(x) \cdot A(x) \cdot V(x) dx. \qquad (2)$$

The western edge of the integration, $x_W$, is set to 13° W, where sea ice does not move throughout year[17], while the eastern edge $x_E$ varies in time depending on sea ice extent. The time series of the monthly mean sea ice volume transport estimate is shown in Fig. 3c, together with volume transport estimates obtained from different combinations of ice thickness and drift speed estimates (Envisat/CS2smos ice thickness, ice drift from the combination of three products). Although the magnitude of peaks in each year differs between the different estimates, they generally agree with the interannual variation of annual volume transport and the drastic reduction in 2018.

**Uncertainty estimates of the sea ice volume transport**. Uncertainty of the monthly mean sea ice volume transport at each zonal segment, $\Delta f$, is calculated by the following formula,

$$\Delta f \simeq |f| \left( \left| \frac{\Delta T}{T} \right| + \left| \frac{\Delta A}{A} \right| + \left| \frac{\Delta V}{V} \right| \right), \qquad (3)$$

where $\Delta T$, $\Delta A$, and $\Delta V$, are uncertainty estimates of monthly mean values of ice thickness, ice concentration, and ice drift speed, respectively. The formula assumes that $\Delta T$, $\Delta A$, and $\Delta V$ are small compared to $T$, $A$, and $V$, respectively, and they are independent variables. Equation (3) takes into account systematic errors in addition to random errors[71]. A constant value of error estimates is used for sea ice concentration (2.37% on a monthly basis, based on the smearing uncertainty of daily sea ice concentration). The uncertainty of sea ice drift speed is given by an empirical error function formula[72], which takes into account an increase of uncertainty in higher drift speed and lower sea ice concentration. The uncertainty of the monthly mean ice thickness $\Delta T$ is estimated based on the accuracies of respective instruments[22,54], with an assumption of no systematic bias and negligible influence of snow depth variation (Table S3). The high sampling rate of ULS/IPS combined with the fast and variable sea ice motion in Fram Strait gives a thickness estimate representing a larger spatial area compared with the slow and stagnant sea ice motion area. The extrapolation of ice thickness outside the zonal range of the mooring array ($x < x_{F14}, x > x_{F11}$) introduces an additional uncertainty associated with the uncertainty of the zonal gradient of thickness,

$$\frac{\Delta T(x)}{|T(x)|} = \frac{\Delta T(x_{F14}) + \Delta \alpha \cdot (x_{F14} - x)}{|T(x_{F14}) + \alpha \cdot (x_{F14} - x)|} \quad (x < x_{F14}), \qquad (4)$$

$$\frac{\Delta T(x)}{|T(x)|} = \frac{\Delta T(x_{F11}) + \Delta \alpha \cdot (x - x_{F11})}{|T(x_{F11}) + \alpha \cdot (x - x_{F11})|} \quad (x > x_{F11}), \qquad (5)$$

where $x_{F14}$ and $x_{F11}$ are locations of the westernmost and the easternmost mooring site F14 and F11, respectively. The standard error of the zonal gradient, $\Delta \alpha$, is given by the standard deviation of the zonal gradient between the mooring sites in the 2010–2018 period (Fig. S10). The uncertainty of the monthly mean ice volume flux is given by

$$\Delta F = \int |T(x) \cdot A(x) \cdot V(x)| \left( \frac{\Delta T(x)}{|T(x)|} + \frac{\Delta A}{|A(x)|} + \frac{\Delta V(A(x), |V(x)|)}{|V(x)|} \right) dx. \qquad (6)$$

The largest source of the uncertainty comes from the zonal extrapolation of ice thickness (49–73%), while ice concentration (drift speed) contributes to 13–24 (13–26) % of the total uncertainty. Uncertainties of respective monthly mean volume flux are shown in Fig. 3c. Uncertainties of annual volume flux in different periods are summarized in Table 1.

**Backward sea ice trajectory analysis**. Backward trajectories of sea ice floes are calculated by the daily sea ice motion vectors from NSIDCv4 (Fig. S11). Pseudo ice floes positioned on the Fram Strait section (8 floes distributed from 0° to 10° W at 0° W, 1.5° W, 3.0° W, 4.0° W, 5.0° W, 6.5° W, 8.0° W, 10.0° W) are advected backwards in time over 4 years using the daily sea ice motion vectors. Ice drift vector at each floe position is calculated by interpolation of daily data at surrounding data points with a Gaussian-type weighting (e-folding scale of 25 km). If no ice drift data are available within 25 km of an ice floe position or sea ice concentration at the floe position is lower than 15%, the trajectory calculation is terminated. The ice concentration at an ice floe position is calculated from an interpolation of daily sea ice concentration (OSI-409/OSI-430) with a Gaussian-type weighting (e-folding scale of 12.5 km). The backward calculations start on the 15th of each month from 1990 to 2018. Trajectory data starting from 2011 to 2018 are combined with sea ice thickness estimates from CS2smos products to examine the evolution of ice thickness along the trajectories.

Uncertainty of daily position of pseudo ice floes is assessed by comparisons with buoy tracks obtained from the International Arctic Buoy Program (IABP)[73]. We used 83 buoy tracks that arrived in Fram Strait in the period from 2000 to 2018 and calculated corresponding pseudo buoy tracks backward in time. The comparisons show that the backward trajectory calculation has ~100 km (230 km) error after 3 months (1 year) calculation back in time and the error can be reasonably

approximated by a linear function of backtracking days (Fig. S12). The error function is used to quantify the uncertainty of physical quantities on the trajectories associated with the position error. A Gaussian kernel, the e-folding scale of which is given by the position error, is applied to estimate the temporal evolution of sea ice thickness and 2 m air temperature along the trajectories.

CS2/CS2smos ice thickness products from three different institutes (AWI CS2smos, Univ. Bristol, and NERC-CPOM) are independently applied to estimate temporal evolutions of ice thickness along the trajectories. Mean positions of the ice floes in each month are calculated by averaging positions of surviving floes on the 15th of a month, and the Gaussian kernel is applied to estimate ice thickness for the position. The temporal evolutions of ice thickness are obtained at monthly interval except for the melting season (May–September) (Fig. 4b and Figs. S1–S6 top panels). The history of atmospheric forcing on ice floes is reconstructed by the evolution of 2 m air temperature along the trajectories. The 2 m air temperature is taken from ERA5 with the same averaging procedure with ice thickness (Fig. 4c).

**Salinity anomaly in Fram Strait in May 2018**. A salinity section in the western part of Fram Strait is analyzed to examine the effect of the ice thickness reduction on the liquid freshwater content in Fram Strait. The salinity anomaly in the EGC in May 2018 (Fig. S13 c) is calculated from the difference between the salinity in May 2018 (Fig. S13 b) and a mean salinity in April/May obtained from five available spring CTD sections (2002, 2005, 2007, 2008, and 2018) (Fig. S13 a). Relative to this mean section, the salinity anomaly in the shallow part of the EGC (0–30 m and 0–70 m) is calculated for 2018, and compared with the salinity anomaly potentially caused by input from the additional sea melt (Table S4). The meltwater input is estimated from the ice thickness anomaly in May 2018 (1.22 m) relative to mean ice thickness in May corresponding to the same 5 years (i.e., 2002, 2005, 2007, 2008, and 2018).

## Data availability

The ice draft data from Fram Strait AOO are available at the Norwegian Polar Data Centre at https://doi.org/10.21334/npolar.2021.5b717274. The altimetry-based sea ice thickness estimates are available at ftp://ftp.awi.de/sea_ice/product/cryosat2_smos/ (AWI CS2smos product), https://data.bas.ac.uk/full-record.php?id=GB/NERC/BAS/PDC/01257 (Univ. Bristol product), http://www.cpom.ucl.ac.uk/csopr/seaice.php (NERC-CPOM product), and https://cds.climate.copernicus.eu/cdsapp#!/dataset/satellite-sea-ice-thickness?tab=form (Envisat). The ice drift data are available at https://nsidc.org/data/NSIDC-0116/versions/4 (NSIDCv4), https://nsidc.org/data/NSIDC-0116/versions/2 (NSIDCv2), ftp://osisaf.met.no/archive/ice/drift_lr/merged/ (OSI-405), https://ads.nipr.ac.jp/vishop/ (KIMURA). The sea ice concentration data are available at ftp://osisaf.met.no/reprocessed/ice/conc/v2p0/ (OSI-409, superseded by OSI-450) and ftp://osisaf.met.no/reprocessed/ice/conc-cont-reproc/v1p2/ (OSI-430). ERA5 reanalysis product is available at https://cds.climate.copernicus.eu/#!/home. The salinity data are available at https://adc.met.no/datasets/10.21343/7jqb-5930 (KV Svalbard cruise 2007), https://adc.met.no/datasets/10.21343/btym-vh89 (KV Svalbard cruise 2008), and https://data.npolar.no/dataset/e3d4f892-2ccc-5b9a-8a6b-8330bc1ec9ee (CTD data of other years).

## Code availability

The backward trajectory code is available on request.

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

## Acknowledgements

This study has been made possible through the long-term observations from the Fram Strait Arctic Outflow Observatory maintained by the Norwegian Polar Institute. This work was supported by the Norwegian Research Council through the FRIPRO program (grant 286971, project FreshArc).

## Author contributions

H.S. conducted all the data analysis in this study, synthesized all inputs, and wrote the paper with significant inputs from L.d.S. L.d.S. organized and implemented the research project of freshwater and sea ice export through Fram Strait. S.G. developed the sea ice part of the project with D.D. O.P. put forward the idea of the backward trajectory analysis. D.D. pointed out the effect of southerly wind and idea for melt rate analysis of the sea ice upstream of the array. L.d.S. contributed to formulate the oceanic effect on sea ice. All the authors contributed to shape and polish ideas, and to finalize the paper.

## Competing interests

The authors declare no competing interests.
