## [Peer Review File · Nature Communications]

Unprecedented decline of Arctic sea ice outflow in 2018Reviewers' Comments:

Reviewer #1:

Remarks to the Author:

Review of 'Unprecedented decline of Arctic sea ice outflow in 2018' by Sumata et al.

Synopsis

The authors examine a decline in observationally estimated sea ice export out from the Arctic Ocean through Fram Strait in 2018, relative to the long-term mean since the 1990s, based on Upward Looking Sonar (ULS) ice thickness measurements at four moorings in Fram Strait combined with remotely sensed observations of ice pack characteristics upstream and atmospheric reanalysis data. They calculate volume export from ice thickness, concentration and drift speed estimates based on the above data. The ice thickness is derived from the ULS ice draft measurements, which recorded significantly smaller magnitudes between September 2017 and August of 2018. In parallel, sea ice concentration was reduced as well during that time, especially along the eastern side of the ice outflow. In addition to these two contributing factors to the decline of ice volume export, the estimated ice drift speed during the first half of this year was ~66% slower relative to the long term-mean, which also contributed to the decline of ice thickness. The analysis of causality of this variability in ice export through Fram Strait involved the time-backward trajectories of ice drift and temporal evolution of ice thickness. The results presented make a convincing case for the anomalously low sea ice export through Fram Strait in 2018. However, their discussion of wider implications is less so. The summary of my main concerns is provided below.

General comments

First and foremost, the authors equate the reduced ice export to the reduction of freshwater export across Fram Strait without making any arguments for it. In fact, they argue themselves that the decreased ice thickness and slower ice drift imply increased melting of sea ice upstream of Fram Strait. In addition, their analysis did not involve the liquid freshwater export at Fram Strait, which could increase relative to the previous years with an intensified ice melt upstream. The results presented do not justify an argument that as a results of decreased ice export more freshwater remained in the Arctic Ocean to increase the stratification there. Neither any results are presented on a diminished impact of the Arctic freshwater export downstream in the northern North Atlantic – see my 3rd point below.

Second, there is no evidence presented or discussed that the melt water due to an anomalous reduction of ice thickness north of Fram Strait could have remained in the Eurasian Basin, especially beyond the winter of 2018. In fact, the ice pack would act to limit or isolate the impact of synoptic scale wind reversals on the upper ocean below, hence the liquid freshwater fluxes at Fram Strait could be much less affected by near surface wind variability.

Third, while the ice export accounts for a large portion of the total (solid+liquid) freshwater flux across Fram Strait it rapidly decreases, due to melt, before reaching Denmark Strait, where the liquid fraction is dominant. This means that regardless of whether ice melts to the north or south of Fram Strait, its relative contribution to the freshwater export into the Irminger and Labrador seas and the North Atlantic does not have to be any different.

Fourth, based on the comments above I don't find any evidence in this paper on a drastic change of the freshwater cycle due to the increased sea ice melt north of and consequently reduced ice export at Fram Strait. In summary, while the observationally estimated ice export at Fram Strait could have been unusually low in 2018, its potential importance to the ocean downstream and European climate remains doubtful.

Specific comments

I.48-50: It's not clear how this paper addresses or support these arguments.

I.53-54: Please provide a reference for this sentence.

I.70: capitalize 'Transpolar Drift'.

I.99-101 & Fig.2: Why not show the ice draft time series since the 1990s, to show the uniqueness of 2018 and interannual variability of a longer time period.

I.109: Please clarify what volume, over what region, is referred here.

I.179-180: This sentence is not clear, a revision is recommended.

I.183: change 'has' to 'have' as this refers to 'anomalies'.

I.202: please clarify if you mean volume, thickness, or else.

I.203-204: while this is an important question, the liquid freshwater fate has not been addressed in the presented analysis.

I.204-207: Again, the fate of liquid freshwater has not been addressed in this paper, nor there is evidence to support a reduction of its export from the Arctic Ocean or into the North Atlantic.

I.216-222: since the sea ice north of Fram Strait was on its way out of the Arctic Ocean, it's hard to imagine that its slowing down impacted the sea ice mass balance of the Arctic or the resulting melt water somehow spread back into the Eurasian Basin. Hence any long-term effects on ice pack or ocean circulation seem unlikely.

I.224-227: it's not clear how the increasing oceanic heat transport into the Arctic Ocean, declining ice thickness and increased ice drift 'have preconditioned the 2018 ice export decline'.

Summary

The first part of this paper, examining the sea ice characteristics upstream of and at Fram Strait to understand the reasons for the unusually low ice export in 2018, is interesting and convincing. The discussion of wider implications of this fact are mostly speculations, which are not based on any facts. I'll leave it to the editor to decide if the major revision of this part would suffice to justify the publication of this paper in Nature Communications.

Reviewer #2:

Remarks to the Author:

Review of "Unprecedented decline of Arctic sea ice outflow in 2018"

By Sumata, de Steur, Gerland, Divine and Pavlova

Reviewer: John Toole, WHOI

Using moored observations of sea ice thickness on the western half of Fram Strait in combination with satellite sea ice products and atmospheric reanalysis fields, the authors document a striking decrease in the export of sea ice from the Arctic in 2018. The result is both interesting and important. I recommend the work be published after minor revision. (I do not need to see the manuscript again prior to publication.)

My main recommendation concerns uncertainty estimates. Although errors and uncertainties are analyzed in detail in the Methods section of the paper, the actual uncertainties are not reported together with the ice transport estimates in the body of the manuscript (appearing only in Table 1 - though even there, the table caption doesn't provide enough information about how to interpret the uncertainty - 95% confidence bounds or ??? Also in that table, standard deviation estimates are provided, but we're not told what they are based on - monthly or annual estimates or ???)

My only other substantive thought references the discussion starting on line 200. The idea that the sea ice reduction was largely caused by the stalling of ice drift south, allowing ocean heat to melt more ice presupposes that there was a sufficient reservoir of ocean heat to support the estimated excess melting and that this heat was able to be brought up to the ice-ocean interface where it could act on the ice. This latter issue is raised around line 219 in the opposite sense - melting increases stratification that can inhibit the vertical flux of heat to the ice-ocean interface. I suggest that these issues be noted in the discussion.

Otherwise, I have some very minor wording suggestions that the authors might wish to address in their revision.

Line 24: perhaps "... just 25% of the 1990s rate."

Line 55: perhaps "...reduction of mean ice thickness in Fram Strait over the last two decades at a rate of approximately 15% per decade."

Lines 62-65: As written, this sentence concerning the pathway of AW felt off track to me. After reading ahead, I suspect the authors are thinking about an ocean heat source that can account for sea ice melting in the region north of Fram Strait. Perhaps this bit of text could be reworked to highlight the large fraction of AW inflow through Fram Strait that extends north before recirculating back south, and remove mention of the AW flow branches that extend east as a boundary current (and thus have less influence on the stratification immediately north of the Strait)

Line 68: perhaps "... East Siberian Sea and to a lesser extent, the Chukchi Sea)..."

Line 71: perhaps "... is the main mechanism transporting ice ..."

Line 75: "Barents Sea maintains this pressure gradient..."

Line 95: perhaps "The observatory has documented long-term variations of the freshwater..."

Line 101: "... ULSs starting in August 2017 and lasting until..."

Line 103: might just delete the phrase "from freezing in autumn to melting in summer."

Line 114: "The minimum fell below the prior minimum recorded..."

Line 115: could use "compounded" in place of "amplified"

Line 179: "... Arctic dipole anomaly with respect to the anomaly centers."

Line 182: "The SLP anomalies that occurred in the 2017-2018 period had more direct impact..."

Line 221: perhaps "Sustained monitoring efforts of both... are necessary to clarify both ... impacts of the event"

Line 227: perhaps just delete the phrase "while the effects of each component needs to be clarified"

Line 243: Four ULS/IPs were zonally aligned

Line 247: were describe in past literature

Line 250: might note around here that the moorings are service annually (or at whatever rate they are replaced)

Line 254: "... erroneous records (e.g. despiking) and sound speed corrections."

Line 274: "...limited to the freezing season..."

Line 393: perhaps "Pseudo ice floes positioned on the Fram Strait section.... are advected backwards in time over 4 years using the daily sea ice motion vectors."

Line 397: drift data ARE available within 25 km of an ice floe position...

Line 418: ...obtained at monthly interval except...

Figure 2: This isn't critical, but I would reverse the way the black arrows are pointing (arrows that mark the time period of each annual average)

Caption of Figure S9: "markers"

Reply to the reviewer's comments.

We sincerely appreciate the Anonymous Reviewer #1 and John Toole for their thorough reviews, comments and suggestions on our manuscript. The reviews helped immensely in the shaping of the manuscript. The suggestions and comments have been closely followed and revisions have been made accordingly.

Best regards,
Hiroshi Sumata on behalf of all authors

In the following the comments from the reviewers are shown in *blue italic fonts* while our replies are in black roman font. The line numbers in our reply (*shown in red*) refer to those in the **tracked-changes manuscript**. References to the papers appeared in the text are provided at the end of main text.

Reviewer #1 (Remarks to the Author):

Review of 'Unprecedented decline of Arctic sea ice outflow in 2018' by Sumata et al.

Synopsis

The authors examine a decline in observationally estimated sea ice export out from the Arctic Ocean through Fram Strait in 2018, relative to the long-term mean since the 1990s, based on Upward Looking Sonar (ULS) ice thickness measurements at four moorings in Fram Strait combined with remotely sensed observations of ice pack characteristics upstream and atmospheric reanalysis data. They calculate volume export from ice thickness, concentration and drift speed estimates based on the above data. The ice thickness is derived from the ULS ice draft measurements, which recorded significantly smaller magnitudes between September 2017 and August of 2018. In parallel, sea ice concentration was reduced as well during that time, especially along the eastern side of the ice outflow. In addition to these two contributing factors to the decline of ice volume export, the estimated ice drift speed during the first half of this year was ~66% slower relative to the long term-mean, which also contributed to the decline of ice thickness. The analysis of causality of this variability in ice export through Fram Strait involved the time-backward trajectories of ice drift and temporal evolution of ice thickness. The results presented make a convincing case for the anomalously low sea ice export through Fram Strait in 2018. However, their discussion of wider implications is less so. The summary of my main concerns is provided below.

General comments

First and foremost, the authors equate the reduced ice export to the reduction of freshwater export across Fram Strait without making any arguments for it. In fact, they argue themselves that the decreased ice thickness and slower ice drift imply increased melting of sea ice upstream of Fram Strait. In addition, their analysis did not involve the liquid freshwater export at Fram Strait, which could increase relative to the previous years with an intensified ice melt upstream. The results presented do not justify an argument that as a results of decreased ice export more freshwater remained in the Arctic Ocean to increase the stratification there. Neither any results are presented on a diminished impact of the Arctic freshwater export downstream in the northern North Atlantic – see my 3rd point below.

Thank you for the comments. We admit that this is an important point that should have been addressed in the manuscript. To address this point, we now included analyses of liquid freshwater

anomalies in the shallow part of the East Greenland Current (EGC) in 2018 and a discussion regarding the effect of sea ice melt water in 2018 (lines 218 – 228, figure S13, table S3). Please find details of these analyses and discussions in our following replies.

Second, there is no evidence presented or discussed that the melt water due to an anomalous reduction of ice thickness north of Fram Strait could have remained in the Eurasian Basin, especially beyond the winter of 2018. In fact, the ice pack would act to limit or isolate the impact of synoptic scale wind reversals on the upper ocean below, hence the liquid freshwater fluxes at Fram Strait could be much less affected by near surface wind variability.

We examined a salinity section in the Fram Strait in spring 2018 and compared it with the potential decrease of salinity caused by the melt water input due to the anomalous reduction of ice thickness in 2018. The salinity section in the Fram Strait in May 2018 shows fresher conditions in the surface layer relative to observations from spring 2002, 2005, 2007 and 2008, concurrent with the sea ice melt in the upstream area (Figure S13a, b). However, the 2018 salinity anomaly in the upper ocean of -0.35 psu (0 – 30 m average, Figure S13c and Table S3), accounts for less than one-third of a potential salinity change due to melt water input associated with the ice thickness anomaly in this period, i.e. 1.22 m ice thickness reduction can cause salinity anomaly of -1.21 psu (0 – 30 m average, Table S3). This suggests that a large part of the melt water remains in the upstream area of Fram Strait, is mixed down, or is exported at least not at the same time. We revised the manuscript accordingly (lines 218 – 225).

Third, while the ice export accounts for a large portion of the total (solid+liquid) freshwater flux across Fram Strait it rapidly decreases, due to melt, before reaching Denmark Strait, where the liquid fraction is dominant. This means that regardless of whether ice melts to the north or south of Fram Strait, its relative contribution to the freshwater export into the Irminger and Labrador seas and the North Atlantic does not have to be any different.

We appreciate the comment. This is also an important point that should have been discussed in the manuscript. To address this point, we now described the difference of impact on downstream areas between sea ice export and liquid freshwater transport by EGC as follows (lines 230 – 239).

“The difference of the fate of sea ice export versus oceanic transport results in different consequences depending on the phase of the freshwater passing through the Fram Strait. A significant portion of sea ice passing through the Fram Strait can drift into the Nordic Seas (Dodd et al., 2009, Dickson et al., 2007). The liquid freshwater transport on the shelf and in the EGC, however, will follow mostly the continental shelf break of Greenland with some loss through the Jan Mayen Current and East Icelandic Current (de Steur et al., 2017, Håvik et al., 2017). Therefore, a liquid freshwater anomaly due to enhanced ice melting upstream of the Fram Strait may propagate further south with the EGC, with a potential consequence for the dense water formation in the North Atlantic Ocean (Ionita et al., 2016). In addition, when sea ice volume carried with the EGC is reduced, a further retreat of the sea ice edge in the Greenland and Iceland Seas occurs (Mayot et al., 2020). This allows for stronger air-sea interactions and water mass modification along the ice edge of the EGC in winter (Våge, K. et al., 2018) and modulates primary production levels in spring (Mayot et al., 2020).”

Fourth, based on the comments above I don't find any evidence in this paper on a drastic change of the freshwater cycle due to the increased sea ice melt north of and consequently reduced ice export at Fram Strait. In summary, while the observationally estimated ice export at Fram Strait could have been unusually low in 2018, its potential importance to the ocean downstream and European climate remains doubtful.

We revised our discussions and wider implications toward more focus on potential impacts on the local hydrography (lines 246 – 264), downstream regional ocean stratification (lines 230 – 236), air-sea interaction processes (lines 236 – 243), and freshwater cycle in a changing climate (lines 266 – 281).

We believe that the revision properly addressed the main concerns of the Reviewer.

Specific comments

l.48-50: It's not clear how this paper addresses or support these arguments.

We modified the sentence to clarify the relation between our result and the general arguments by addressing phase of freshwater (line 48).

l.53-54: Please provide a reference for this sentence.

We provided two references for this sentence, Serreze et al. (2006) and Haine et al., (2015) (line 55).

l.70: capitalize 'Transpolar Drift'.

We capitalized 'Transpolar Drift' (line 75).

l.99-101 & Fig.2: Why not show the ice draft time series since the 1990s, to show the uniqueness of 2018 and interannual variability of a longer time period.

Figure 2 is prepared to describe the sudden drop of sea ice thickness occurred in September 2017 and lasted until August 2018, in addition to the relation between thickness and concentration in this period (lines 104 - 114). The figure requires high temporal resolution for this purpose, which is not possible if the figure covers the entire monitoring period (1990 - 2018).

To address the reviewers concern, we provided longer time series (2003 – 2018) of daily mean ice thickness and effective thickness in Figure S14 and revised the caption of Figure 2 accordingly. The thickness record covering the entire period (1990 – 2018) is shown in Figure 3a, which clearly describes the uniqueness of 2018 and interannual variability of a longer time period. Since we applied a constant conversion ratio from draft to thickness (lines 303 - 304), Figure 3a can be interpreted as time series of ice draft by multiplying with 0.88.

l.109: Please clarify what volume, over what region, is referred here.

Thank you for pointing out our insufficient description. The reduction of annual mean thickness (0.74 m) and ice volume (55%) are the average of the four mooring sites covering the transect from 3° W to 6.5° W at 78.8° N. We revised the main text accordingly (lines 113 – 114).

l.179-180: This sentence is not clear, a revision is recommended.

We revised the sentence as follows, "The anomaly patterns are different from the 'Arctic dipole anomaly' with respect to the anomaly centers." (lines 185 – 186).

l.183: change 'has' to 'have' as this referrers to 'anomalies'.

We revised the sentence, though we used past tense, 'had'. (line 189)

l.202: please clarify if you mean volume, thickness, or else.

Thank you for pointing out the ambiguity.

This sentence refers to sea ice volume export. We revised the sentence accordingly. (line 208)

l.203-204: while this is an important question, the liquid freshwater fate has not been addressed in the presented analysis.

We provided liquid freshwater analysis in 2018 to address the main concern of the reviewer #1. (lines 218 – 225 in the main text, Figure S13, Table S3 and corresponding description of methods (lines 479 – 489) and data used (lines 506 – 508).

l.204-207: Again, the fate of liquid freshwater has not been addressed in this paper, nor there is evidence to support a reduction of its export from the Arctic Ocean or into the North Atlantic.

Please find our reply above.

l.216-222: since the sea ice north of Fram Strait was on its way out of the Arctic Ocean, it's hard to imagine that its slowing down impacted the sea ice mass balance of the Arctic or the resulting melt water somehow spread back into the Eurasian Basin. Hence any long-term effects on ice pack or ocean circulation seem unlikely.

Thank you for this comment.

For individual years, sea ice volume export through Fram Strait can very well have contributed to anomalies in the overall sea ice volume in the Arctic Basin. For example, the extreme sea ice area minima in 2007 and 2012 was preceded by an enhanced sea ice volume export anomaly in the months before the September minimum (Spreen et al., 2020). As the reviewer pointed out, however, it is still unclear to what extent the changes of the export has contributed to the overall sea ice volume decline in the Arctic Basin on the long term.

To address the reviewer's concern, we removed the sentence and revised our discussion with a focus on the local sea ice retreat and hydrographic changes in the northeast of the Fram Strait as follows,

“It is also important to note the consequences of the anomalous thickness reduction and the southerly wind events on the local hydrography, water mass modification and biological processes north of the Fram Strait. A recent study reported that anomalous reduction of sea ice concentration and extreme deep mixed layers occurred northeast of Fram Strait in the 2017/2018 winter (Athanasé et al., 2020). Another study pointed out that the anomalous ocean stratification in 2018 affected local biogeochemical cycles north of the Fram Strait (von Appen et al., 2021). The anomalous ice melt reported here can have contributed to the reduction of sea ice cover in this area, while the southerly winds can have advected the melt water eastward by Ekman transport and contributed to the observed fresher surface layer and shallower warm AW layer observed in the western Eurasian basin in winter 2017 (Athanasé et al., 2019). The anomalous thickness reduction, together with the observed hydrographic anomalies in the northeast of Fram Strait, might be a leading signal of further retreat of marginal ice zones and concurrent northward shift of water mass modification area in the Atlantic sector of the Arctic in the coming years to decades.” (lines 246 – 256)

l.224-227: it's not clear how the increasing oceanic heat transport into the Arctic Ocean, declining ice thickness and increased ice drift 'have preconditioned the 2018 ice export decline'.

Thank you for pointing the ambiguity of our discussion. We revised the sentence so as to clarify how each component can precondition the 2018 ice export decline as follows,

“More heat transport into the Arctic Ocean (Polyakov et al., 2017, Tsubouchi et al., 2021) might result in an increase of ocean heat content available for melting sea ice, a decline of basin-wide ice

thickness (Kwok, 2018) making ice floes more vulnerable to environmental changes, an increase of mobility of sea ice in a number of regions (Spreen et al., 2011) making sea ice more sensitive to local wind forcing; all these might have preconditioned the 2018 ice export decline.” (lines 267 – 270)

Summary

The first part of this paper, examining the sea ice characteristics upstream of and at Fram Strait to understand the reasons for the unusually low ice export in 2018, is interesting and convincing. The discussion of wider implications of this fact are mostly speculations, which are not based on any facts. I'll leave it to the editor to decide if the major revision of this part would suffice to justify the publication of this paper in Nature Communications.

Here we again appreciate all the comments, criticisms and suggestions from the reviewer #1, that significantly helped us to strengthen discussions and implications of this work. We believe that the revised manuscript satisfactorily addressed all the concerns of reviewer #1.

Reviewer #2 (Remarks to the Author):

*Review of “Unprecedented decline of Arctic sea ice outflow in 2018”
By Sumata, de Steur, Gerland, Divine and Pavlova
Reviewer: John Toole, WHOI*

Using moored observations of sea ice thickness on the western half of Fram Strait in combination with satellite sea ice products and atmospheric reanalysis fields, the authors document a striking decrease in the export of sea ice from the Arctic in 2018. The result is both interesting and important. I recommend the work be published after minor revision. (I do not need to see the manuscript again prior to publication.)

We sincerely appreciate John Toole for the review and suggestions/corrections on the manuscript.

My main recommendation concerns uncertainty estimates. Although errors and uncertainties are analyzed in detail in the Methods section of the paper, the actual uncertainties are not reported together with the ice transport estimates in the body of the manuscript (appearing only in Table 1 - though even there, the table caption doesn't provide enough information about how to interpret the uncertainty - 95% confidence bounds or ??? Also in that table, standard deviation estimates are provided, but we're not told what they are based on - monthly or annual estimates or ???)

Thank you for the suggestion. We implemented uncertainty of the monthly mean ice volume transport in Figure 3c. To clearly show the time-varying uncertainty, aspect ratio of the plot has been changed. We also implemented caption into the Table 1, describing that the uncertainties and standard deviations refer to those of the annual mean volume flux in each period and in 2018 and that how they were derived. The main text was also revised accordingly (lines 439 – 441).

My only other substantive thought references the discussion starting on line 200. The idea that the sea ice reduction was largely caused by the stalling of ice drift south, allowing ocean heat to melt more ice presupposes that there was a sufficient reservoir of ocean heat to support the estimated excess melting and

that this heat was able to be brought up to the ice-ocean interface where it could act on the ice. This latter issue is raised around line 219 in the opposite sense – melting increases stratification that can inhibit the vertical flux of heat to the ice-ocean interface. I suggest that these issues be noted in the discussion.

Thank you for the comments. We revised the discussion about the effect of sea ice melt on the Arctic hydrography by referring observed hydrographic anomalies in 2017/2018 as follows,

“It is also important to note the consequences of the anomalous thickness reduction and the southerly wind events on the local hydrography, water mass modification and biological processes north of the Fram Strait. A recent study reported that anomalous reduction of sea ice concentration and extreme deep mixed layers occurred northeast of Fram Strait in the 2017/2018 winter (Athanasé et al., 2020). Another study pointed out that the anomalous ocean stratification in 2018 affected local biogeochemical cycles north of the Fram Strait (von Appen et al., 2021). The anomalous ice melt reported here can have contributed to the reduction of sea ice cover in this area, while the southerly winds can have advected the melt water eastward by Ekman transport and contributed to the observed fresher surface layer and shallower warm AW layer observed in the western Eurasian basin in winter 2017 (Athanasé et al., 2019). The anomalous thickness reduction, together with the observed hydrographic anomalies in the northeast of Fram Strait, might be a leading signal of further retreat of marginal ice zones and concurrent northward shift of water mass modification area in the Atlantic sector of the Arctic in the coming years to decades.” (lines 246 – 256)

Otherwise, I have some very minor wording suggestions that the authors might wish to address in their revision.

Line 24: perhaps “... just 25% of the 1990s rate.”

Thank you for the suggestion, but removed this sentence to follow the word limit of the abstract.

Line 55: perhaps “...reduction of mean ice thickness in Fram Strait over the last two decades at a rate of approximately 15% per decade.”

The manuscript was revised as suggested. (lines 56 – 57)

Lines 62-65: As written, this sentence concerning the pathway of AW felt off track to me. After reading ahead, I suspect the authors are thinking about an ocean heat source that can account for sea ice melting in the region north of Fram Strait. Perhaps this bit of text could be reworked to highlight the large fraction of AW inflow through Fram Strait that extends north before recirculating back south, and remove mention of the AW flow branches that extend east as a boundary current (and thus have less influence on the stratification immediately north of the Strait)

Thank you for suggestion. We revised the sentences as follows,

“Within and just north of the strait, the EGC meets returning warm and saline Atlantic Water (AW) that circulates westward following several pathways between that are typically associated with large eddy variability (Hattermann et al, 2016, von Appen et al., 2016). The recirculating AW meets and may subduct under the fresher Polar Water, and flow southward again along the Polar front with the EGC. In 2018, AW was found as far north as 82.8°N west of the Yermak Plateau (Athanasé et al., 2019).” (lines 63 - 67)

Line 68: perhaps “... East Siberian Sea and to a lesser extent, the Chukchi Sea)...”

We revised the text as suggested. (line 72).

Line 71: perhaps "... is the main mechanism transporting ice ..."

We revised the text as suggested. (line 76)

Line 75: "Barents Sea maintains this pressure gradient..."

We revised the text as suggested. (line 79)

Line 95: perhaps "The observatory has documented long-term variations of the freshwater..."

The text was revised accordingly. (line 101).

Line 101: "... ULSs starting in August 2017 and lasting until..."

We revised the text as suggested. (line 106)

Line 103: might just delete the phrase "from freezing in autumn to melting in summer."

We revised the text as suggested. (line 108)

Line 114: "The minimum fell below the prior minimum recorded..."

We revised the text as suggested. (line 119)

Line 115: could use "compounded" in place of "amplified"

We revised the text as suggested. (line 121)

Line 179: "... Arctic dipole anomaly with respect to the anomaly centers."

We revised the text as suggested. (lines 185-186)

Line 182: "The SLP anomalies that occurred in the 2017-2018 period had more direct impact..."

We revised the text as suggested. (line 189)

Line 221: perhaps "Sustained monitoring efforts of both... are necessary to clarify both ... impacts of the event"

We revised the text as suggested. (lines 262 - 263)

Line 227: perhaps just delete the phrase "while the effects of each component needs to be clarified"

We revised the text as suggested. (lines 272 - 273)

Line 243: Four ULS/IPSS were zonally aligned

We revised the text as suggested. (line 291)

Line 247: were describe in past literature

We revised the text as suggested. (line 295)

Line 250: might note around here that the moorings are service annually (or at whatever rate they are replaced)

We noted this in line 294.

Line 254: "... erroneous records (e.g. despiking) and sound speed corrections."

We revised the text as suggested. (line 303)

Line 274: "...limited to the freezing season..."

We revised the text as suggested. (line 322)

Line 393: perhaps "Pseudo ice floes positioned on the Fram Strait section.... are advected backwards in time over 4 years using the daily sea ice motion vectors."

We revised the text as suggested. (lines 447 - 449)

Line 397: drift data ARE available within 25 km of an ice floe position...

We revised the text as suggested. (lines 451 - 452)

Line 418: ...obtained at monthly interval except...

We revised the text as suggested. (line 473)

Figure 2: This isn't critical, but I would reverse the way the black arrows are pointing (arrows that mark the time period of each annual average)

We revised Figure 2 as suggested.

Caption of Figure S9: "markers"

We revised the caption accordingly.

Thank you again for the thorough reviews on our manuscript.

References

1. Athanase, M. et al. New hydrographic measurements of the upper Arctic western Eurasian Basin in 2017 reveal fresher mixed layer and shallower warm layer than 2005–2012 climatology. *J. Geophys. Res. Oceans* **124**, 1091–1114 (2019).

2. Athanase, M., et al. Atlantic water modification north of Svalbard in the Mercator physical system from 2007 to 2020. *J. Geophys. Res.: Oceans*, 125, e2020JC016463. <https://doi.org/10.1029/2020JC016463> (2020).
3. de Steur, et al., Liquid freshwater transport estimates from the East Greenland Current based on continuous measurements north of Denmark Strait, *J. Geophys. Res. Oceans*, 122, 93–109, doi:10.1002/2016JC012106 (2017).
4. Dickson et al., Current estimates of freshwater flux through Arctic and subarctic seas, *Progress in Oceanography*, 73, 210 – 230, doi:10.1016/j.pocean.2006.12.003 (2007).
5. Dodd, P. A. et al., Sources and fate of freshwater exported in the East Greenland Current, *Geophys. Res. Lett.*, 36, L19608, doi:10.1029/2009GL039663 (2009).
6. Haine, T. W. N. et al. Arctic freshwater export: Status, mechanisms, and prospects. *Global Planet. Change* **125**, 13– 35 (2015).
7. Hattermann, T., Isachsen, P. E., von Appen, W.-J., Albretsen, J. and Sundfjord, A. Eddy-driven recirculation of Atlantic Water in Fram Strait. *Geophys. Res. Lett.* **43**, 3406–3414 (2016).
8. Håvik, L. et al., Evolution of the East Greenland Current from Fram Strait to Denmark Strait: Synoptic measurements from summer 2012, *J. Geophys. Res. Oceans*, 122, 1974 – 1994, doi:10.1002/2016JC012228 (2017).
9. Ionita, M. et al., Linkages between atmospheric blocking, sea ice export through Fram Strait and the Atlantic Meridional Overturning Circulation. *Scientific Reports*, 6, 32881, <https://doi.org/10.1038/srep32881> (2016).
10. Kwok, R. Arctic sea ice thickness, volume, and multiyear ice coverage: losses and coupled variability (1958 – 2018). *Environ. Res. Lett.* **13**, 105005 (2018).
11. Mayot, N. et al., Spring export of Arctic sea ice influences phytoplankton production in the Greenland Sea, *J. Geophys. Res. Oceans*, 125, e2019JC015799, doi:10.1029/2019JC015799 (2020).
12. Polyakov, I. V. et al. Greater role for Atlantic inflows on sea-ice loss in the Eurasian Basin of the Arctic Ocean. *Science* **356**, 285-291 (2017).
13. Serreze, M. C. et al. The large-scale freshwater cycle of the Arctic. *J. Geophys. Res.* **111**, C11010 (2006).
14. Spreen, G., Kwok, R., Menemenlis, D. Trends in Arctic sea ice drift and role of wind forcing: 1992 – 2009. *Geophys. Res. Lett.* **38**, L19501, doi:10.1029/2011GL048970 (2011).
15. Spreen, G. et al. Arctic sea ice volume export through Fram Strait from 1992 to 2014. *J. Geophys. Res. Oceans* **125**, e2019JC016039 (2020).
16. Tsubouchi, T. et al. Increased ocean heat transport into the Nordic Seas and Arctic Ocean over the period 1993–2016. *Nat. Clim. Chang.* **11**, 21–26 (2021).
17. Våge, K. et al., Ocean convection linked to the recent ice edge retreat along east Greenland, *nature communications*, doi:10.1038/s41467-018-03468-6 (2018).
18. von Appen, W.-J., Schauer, U., Hattermann, T. and Beszczynska-Möller, A. Seasonal Cycle of Mesoscale Instability of the West Spitsbergen Current. *J. Phys. Oceanogr.* **46**, 1231 – 1254 (2016).

Reviewers' Comments:

Reviewer #2:

Remarks to the Author:

I believe the authors have satisfactorily addressed my criticisms of their previous submission and believe the work is suitable for publication in Nature Communications.

Reply to the review comments.

In the following the comments from the reviewers are shown in *blue italic fonts* while our replies are in black roman font.

Reviewer #2 (Remarks to the Author):

I believe the authors have satisfactorily addressed my criticisms of their previous submission and believe the work is suitable for publication in Nature Communications.

We sincerely appreciate the anonymous reviewer for taking time for the second round of the review.

Here we again appreciate the anonymous reviewer and John Toole for reviewing our manuscript.

Best regards,
Hiroshi Sumata
On behalf of all authors